# Dental Lesion Segmentation Using an Improved ICNet Network with Attention

**DOI:** 10.3390/mi13111920

**Published:** 2022-11-07

**Authors:** Tian Ma, Xinlei Zhou, Jiayi Yang, Boyang Meng, Jiali Qian, Jiehui Zhang, Gang Ge

**Affiliations:** 1College of Computer Science and Technology, Xi’an University of Science and Technology, Xi’an 710054, China; 2Department of Electrical and Computer Engineering, National University of Singapore, Singapore 117583, Singapore

**Keywords:** attention model, asymmetric convolution, ICNet, tooth lesions

## Abstract

Precise segmentation of tooth lesions is critical to creation of an intelligent tooth lesion detection system. As a solution to the problem that tooth lesions are similar to normal tooth tissues and difficult to segment, an improved segmentation method of the image cascade network (ICNet) network is proposed to segment various lesion types, such as calculus, gingivitis, and tartar. First, the ICNet network model is used to achieve real-time segmentation of lesions. Second, the Convolutional Block Attention Module (CBAM) is integrated into the ICNet network structure, and large-size convolutions in the spatial attention module are replaced with layered dilated convolutions to enhance the relevant features while suppressing useless features and solve the problem of inaccurate lesion segmentations. Finally, part of the convolution in the network model is replaced with an asymmetric convolution to reduce the calculations added by the attention module. Experimental results show that compared with Fully Convolutional Networks (FCN), U-Net, SegNet, and other segmentation algorithms, our method has a significant improvement in the segmentation effect, and the image processing frequency is higher, which satisfies the real-time requirements of tooth lesion segmentation accuracy.

## 1. Introduction

Continuous advancements in the field of computer vision have propelled online intelligent diagnosis and treatment system research. As human living conditions have improved, the interest in dental lesions has increased, and the impact of the epidemic has made hospital diagnoses inconvenient. The following problems occur in clinical settings: (1) Some lesions are similar to the tooth structure, and doctors are prone to miss or misdiagnose the problem. (2) With an increased number of consultations, reviewing numerous films has increased the workload for doctors, resulting in slower review times and a lack of timely feedback to patients. Because of an unequal distribution of medical resources, patients in remote areas lack access to in-depth treatment options. As a result, the online clinic industry continues to emerge to meet the additional needs. The dental lesion identification system can play a role in pre-diagnosis and auxiliary diagnoses so that patients with less severe problems can save time in seeing a doctor, and patients with severe diseases can be diagnosed thoroughly and can obtain their dental lesion diagnosis results anytime and anywhere. This reduces the number of on-site diagnoses necessary during the epidemic and prevents spreading of the COVID-19 virus. The study of real-time segmentation algorithms for dental lesions has become the key to developing intelligent dental lesion detection systems.

Dental diagnosis technology primarily enhances or segments X-ray films and optical coherence tomography (OCT) images to assist doctors in their diagnoses. Lee et al. [1] proposed vertical intensity transform function (VIFT) to solve the problem of lessening the illumination-based differences in tooth grayscales, then used the K-means algorithm and Markov random field to specify the detection range and finally segmented the candidate block of dental calculus. However, the accuracy of these traditional methods for specific dental diagnosis tasks still has room for improvement, and for a single image with multiple lesions, traditional methods cannot identify and segment multiple categories.

At present, many scholars are also trying to use deep learning methods for dental diagnoses. Kreis [2] used Convolutional Neural Network (CNN) to detect periodontal bone loss (PBL) on panoramic dental X-rays. Casalegno et al. [3] used near-infrared light transmission (NILT) images for caries segmentation. Jae-Hong Lee [4] evaluated the effectiveness of the deep CNN algorithm in detecting and diagnosing caries on X-rays of the periapical period. Yu [5] and others also evaluated the performance of the CNN in the classification of bones by lateral head measurement. Recently, Wen et al. [6] used deep learning methods to detect tooth lesions. They built a multitask network structure, and the model was composed of three subnets: FNet (feature extraction subnet), LNet (location subnet), and CNet (classification subnet). This work mainly detected dental calculus, gingivitis, and soft deposits. They identified calculus and gingivitis with different color candidate frames, while for soft deposits, only an image-level classification was performed. The problem with this work is that the shape of the tooth lesion is irregular. The rectangular labeling method learns the characteristics of the surrounding normal tissues by using the network, and the detection result range is larger than the actual pixel range of the lesion. In addition, the LNet subnet in the network model is a two-stage positioning network, so the real-time nature of the model is a problem. Different from previous work, we used an intraoral camera to collect RGB images of dental lesions and used deep learning methods to segment multiple lesions at the pixel level to more accurately detect the scope of the lesions.

Many diagnostic methods are based on currently popular 3D imaging techniques such as CT, 3DMD, and others [7]. Therefore, 3D point cloud data segmentation is also one of the hot research topics. Karatas et al. introduce basic 3D image segmentation methods and summarize the current status of 3D imaging techniques and evaluate their application in orthodontics. To address the sparsity of point cloud data, Graham et al. proposed new sparse convolutional operations SSCNs [8] to handle sparse data more efficiently. Liu et al. proposed a convolutional BEACon network [9] with embedded attentional boundaries for point cloud instance segmentation, which combines geometry and color into attentional weights based on how humans perceive geometry and color for object recognition motives. SMU-Net [10] uses saliency mapping to guide the primary and secondary networks to learn foreground saliency and background saliency representations, respectively, to obtain good segmentation results, but its effectiveness for edge segmentation of fuzzy lesions on small-scale datasets needs to be improved.

Deep learning uses end-to-end training to predict complex models and is able to accomplish lesion segmentation in complex scenarios. Long et al. [11] proposed a fully convolutional network based on CNN, which, for the first time, achieved a pixel-level classification. They cleverly used convolutional layers to replace fully connected layers that contained complex calculations and used deconvolutions to restore the original sizes after convolutions. Subsequently, encoder–decoder was widely used in semantic segmentation. The proposed SegNet [12] network is similar to FCN, with the difference that the location of the maximum value is recorded during the maximum pooling operation in the decoding operation, and then a nonlinear upsampling is achieved by the corresponding pooling index during decoding. A sparse feature map is obtained after upsampling, and then a dense feature map is obtained by ordinary convolution, and the upsampling is repeated. This reduces the amount of computation in the encoding phase. The U-Net [13] proposed by Ronneberger et al. is an improvement on the encoder–decoder architecture, which by connecting the corresponding layers of the encoder and decoder, the low-level and high-level features are merged to bridge the gap.

In the actual segmentation scenario, we have to pursue real-time performance while ensuring accuracy. ENet [14] believes that the decoding structure is only used for the output of upsampling coding and is only used for fine-tuning the edge details, so it does not need to be particularly deep. In addition, the full convolution process is very time-consuming, so ENet only uses one layer of full convolution and makes use of fewer parameters and obtains a faster speed. However, ENet guarantees real-time performance while giving up a certain accuracy rate, resulting in lower segmentation accuracy. ICNet [15] uses the pyramid pooling module of PSPNet [16] to fuse multiscale context information and divide the network structure into three branches: low resolution, medium resolution and high resolution. It uses low resolution to complete the semantic segmentation, and a high-resolution strategy to refine the segmentation results improves the model’s segmentation accuracy. In addition, its use of cascading labels to guide the training of each branch speeds up the model’s convergence and prediction and improves real-time performance.

The above segmentation network has good performance in the segmentation of remote sensing images [17], street view images [18], and lesion images [19], but it does not have an application in dental lesions thus far. We found that there is oversegmentation and undersegmentation when testing with segmentation algorithms such as FCN, SegNet, ENet, etc. The analysis believes that tooth lesions and normal tooth tissues are similar in texture and color and have similar features, which makes it difficult to segment tooth lesions and their edges correctly. In addition, the segmentation of dental lesions needs to ensure accuracy in a reasonable time, thus meeting the needs of the diagnostic equipment and massive amounts of data. In this article, we propose to integrate the attention mechanism into the ICNet network to solve the above two problems. The salient points and contributions of this paper are as follows:The self-built dental lesion dataset included four types of lesions: calculus, gingivitis, tartar, and worn surfaces and was preprocessed with the ACE color equalization algorithm for overexposed images caused by light sources.The lightweight Convolutional Block Attention Module (CBAM) attention module is integrated into the low and middle branches of the image cascade network (ICNet) network so that the high-resolution branches can better guide the features of the low and middle branches, and the large-size convolution of spatial attention uses stacked hollow volumes for product replacement.The regular convolution in the low- and medium-resolution branches are replaced with asymmetric convolutions to reduce the computational effort.

## 2. Related Work

In this section, we introduce the semantic segmentation architecture and model related to our method. These architectures and models are widely used in image segmentation tasks.

### 2.1. Encoder–Decoder Network

Our segmentation network is based on the encoder-decoder network structure. In 2015, the FCN [11] proposed by Long et al. used a convolution layer to replace the complete convolution layer of the entire network and used the deconvolutional layer for upsampling to restore the segmentation results. This fully convolutional network is called an encoder–decoder network [20,21,22]. Since then, most image segmentation networks have adopted the form of codec networks. Based on FCN, U-Net [13] built a more complex decoder, adding compensation at the corresponding level to compensate for local information. SegNet [12] is a further extension of U-Net, which implements the maximum pixel pool operation in the encoder model and reduces the amount of calculation in the decoding stage. ICNet [15] first changes the input image size to one-half and one-quarter of the original image and combines the original image to form the input image, which is input into the low-, medium-, and high-resolution branches. There are many network layers at the low and medium resolutions, but the image resolution is low, which saves calculation time. Although the high-resolution branch has a large image, the number of input network layers is small, resulting in a relatively small time overhead, thus achieving the real-time goal. The feature maps extracted from each branch are fused through the CFF [15] module, and finally, the segmentation results are obtained through decoding.

### 2.2. Attention Mechanism

A limitation of CNN is that it is difficult to effectively learn the global information, and the segmentation of the details of an image is not perfect. In recent years, researchers have integrated feature fusion and attention mechanisms [22,23,24,25] into network models to increase the network’s ability to learn global features to better segment the details. PspNet is proposed through the pyramid pooling module, which aggregates contextual information based on different regions and extends pixel-level features to a specially designed global pyramid pooling. Local and global clues work together to make the final prediction more reliable. In recent years, a large number of studies have proven that introducing the attention module into the network model can effectively improve performance. SA-UNet [26] introduced a spatial attention module SE [27], which can infer attention maps along the spatial dimension and multiply the attention maps with the input feature maps for adaptive feature refinement. The effect of this work on retinal vessel segmentation exceeds the original U-Net of Ronneberger et al. [24] proposed an attention gate structure. The AG module is connected to the end of each hop connection, and the attention mechanism is implemented for the extracted features. In our proposed model, we use the convolution block attention module CBAM, which takes the input image and applies the attention sequence to the channel and then applies it to the spatial dimension. The result of CBAM is a weighted feature map that takes into account the channels and spatial regions of the input image.

### 2.3. Multiple Forms of Convolution

Many studies have proposed various forms of convolutions to improve the performance of the segmentation network. Szegedy et al. [28] first proposed the concept of a 1 × 1 convolution, and its main function was to reduce the dimensionality and save calculation costs. The 1 × 1 convolution is not different from a conventional convolution in terms of the convolution method; the main difference lies in the application scenario and function. They also proposed an asymmetric convolution; that is, the convolution of n*n can be replaced by the convolution of 1 × n followed by n × 1 so that the effect obtained is the same as that of a conventional convolution and the calculation amounts are reduced. A depth separable convolution [29] is an innovation based on a 1 × 1 convolution, and it includes two parts, a deep convolution and 1 × 1 convolution. The purpose of a convolution is to convolve each of the inputs separately using a convolution kernel to convolve it; that is, the channels are separated and then combined. Koltun et al. [30] first used a dilated convolution for image segmentation. A dilated convolution is the process of expanding the convolution kernel by adding some spaces between the elements of the convolution kernel, which increases the receptive field without increasing the number of parameters. The DeepLabV2 algorithm proposed by Papandreou et al. [31] uses a dilated convolution to extract features. Many studies have used a dilated convolution [31,32] to replace a conventional convolution.

## 3. Methods

### 3.1. Adding the CBAM Attention Module to Low- and Medium-Resolution Branches

The network framework in this paper is mainly an ICNet network, which consists of three network branches: low-, medium-, and high-resolution branches, as shown in Figure 1. In our designed network, the low-resolution network is designed according to the first 11 convolutional blocks of Resnet50, using dilated convolution to enhance the feature perception field and asymmetric convolution to reduce the computational effort. The medium-resolution branch is designed according to the first 5 convolution blocks of Resnet50, using asymmetric convolution in the last two convolutions, and both branch network heads include a 7 × 7 convolution and maximum average pooling. The high-resolution branch is designed as a lightweight network, which consists of three convolution blocks, including 3 × 3 convolution, Relu activation function, BN layer, 1 × 1 convolution, and Relu activation function, in that order.

There is a morphological similarity between dental lesions and normal teeth, and the edges are difficult to distinguish. In addition, although ICNet retains most of the semantic information when performing feature extraction in low-resolution branches, there is a loss of detail and edge information, which results in an unsatisfactory segmentation accuracy. In this paper, the CBAM module is added to the low-resolution branch, as shown in Figure 1. Through the serial combination of channel and spatial attention, important features can be enhanced, and unimportant features can be suppressed, thereby improving the performance of the network and improving the model’s ability to learn details.

CBAM applies attention to both channel and spatial dimensions, as shown in Figure 2. CBAM, like the SE module, can be embedded in most current mainstream networks. It can improve the feature extraction ability of the network model without significantly increasing the number of calculations and parameters.

CBAM includes two pieces of content, the channel attention module and spatial attention module; that is, the channel attention CAM and spatial attention SAM.

The network embedded with the CBAM mechanism first performs global pooling and maximum pooling on the constitutionally generated feature maps F using channel attention mapping. The pooling result is connected to the multilayer perceptron for the addition operation, and the channel weight coefficient Mc is generated through the sigmoid activation function. Finally, this weighting coefficient is multiplied by the original feature map F to obtain the feature map F’ after the channel weighting adjustment as shown in Figure 3. The channel attention mapping process is shown in Equation (1).
(1)$$M_{c}=\delta \left(\mathrm{MLP}\left(\mathrm{AvgPool}\left(F\right)\right)+\mathrm{MLP}\left(\mathrm{MaxPool}\left(F\right)\right)\right)$$

In Equation (1), MLP stands for the multilayer perceptron; δ is the activation function.

After that, the feature map generated by the channel attention is sent to the spatial attention for processing. Spatial attention mapping performs a serial connection of global maximum pooling and average pooling on the weighted feature map F’, uses convolution to reduce the dimensionality into a single-channel feature map, and uses the sigmoid function to activate the spatial feature matrix Ms. The weight matrix and the feature map F are subjected to the dot multiplication operation to obtain the final required spatial feature map F”, and the spatial attention mapping process is shown in Equation (2):(2)$$M_{s}\left(F\right)=\delta \left(f^{7\times 7}\left\{\mathrm{AvgPool}\left(F\right);\mathrm{MaxPool}\left(F\right)\right\}\right)$$

In the formula, f^7×7^ represents that the convolution kernel is a 7 × 7 convolution layer; δ is the activation function; and; represents the serial connection.

In this paper, we replace the 7 × 7 large convolution kernel in the spatial attention module with two 3 × 3 dilated convolutions with a dilated rate of 2, reducing the number of parameters in the spatial attention module. The modified CBAM model is shown in Figure 4, where X and Y represent the input and output feature matrices, respectively, and C, W, and H and C^l^, W^l^, and H^l^ represent the three-dimensional information of X and Y, respectively.

To suppress the influence of the useless features on the model, the CBAM attention model is connected to the outside convolution of the last three convolution blocks in the one-quarter resolution branch and the last two convolution blocks in the one-half resolution to improve the segmentation accuracy. The convolution block in front of the low- and medium-resolution branches mainly extracts image features, and then the attention layer is used to enhance the feature extraction so that the obtained features are more accurate, and the high-resolution branch can better guide us in low resolution and medium resolution to better achieve the segmentation effect.

Finally, the feature maps generated by the three branches are fused through the CFF module, as show in Figure 5.

The CFF module has three inputs, the feature maps F1, F2, and label. For F1, it is upsampled twice to make it the same size as F2, and then the features of F1 are refined by convolution with a hole of size 3 × 3 and a dilation rate of 2. For F2, it is convolved by 1 × 1 to make it the same number of channels as F1, and then the features are normalized using the BN layer. This is then added to the F1 features obtained above to obtain F2′. To enhance the learning of feature F1, the upsampled features of 1 are guided using an auxiliary label, optimizing the loss. Where, for the first CFF module, F1 and F2 are the features obtained from the low- and medium-resolution branches, respectively.

### 3.2. Asymmetric Convolution Replaces Regular Convolution

A large convolution kernel can create a larger receptive field, but it also means there are more parameters. An asymmetric convolution can greatly reduce the calculation amounts in the convolution stage without reducing the accuracy, which reduces the size of the model and improves the real-time segmentation of the model. This article replaces part of the regular convolution with an asymmetric convolution to further reduce the quantity of ICNet calculations.

Adding the CBAM attention module to the ICNet network results in a small increase in the calculations of the model. Compared with conventional convolution, asymmetric convolution can greatly reduce the amount of calculation in the convolution stage without losing accuracy. First, an n × 1 convolution is performed, then a 1 × n convolution is performed, as shown in Figure 6, which is consistent with the result of directly performing an n × n convolution, but the scale of the multiplication operation changes from n × n to 2 × n; so, the larger the n, the more obvious the effect of asymmetric convolution in reducing the calculation amounts. In this paper, the one-half and one-quarter resolution branches replace the 3 × 3 convolution with a 3 × 1 convolution, and then a 1 × 3 convolution is joined, as shown in Figure 6, thereby reducing the calculation amounts.

## 4. Experiment

### 4.1. Datasets

#### 4.1.1. Data Collection

An intraoral camera was used to collect images of dental calculus, gingivitis, tartar, and worn surfaces. Two hundred images of dental lesions were obtained by shooting from three angles: the exterior, interior, and top views. The number of occurrences of various lesions in the collected images are shown in Table 1.

#### 4.1.2. Data Augmentation

To expand the difference between the samples and ensure the generalization ability of the later model training, randomly crop the collected dental lesion pictures and restore the original size and flip it over, then adjust the contrast, brightness, saturation, etc. Then, the original tooth lesion image is scaled to 512 × 512 pixels according to the principle of proportional invariance, the distorted image is removed by manual screening, and 400 dental lesion images are selected as the original dataset. Figure 7 shows that dataset example and Figure 8 shows that the number of various lesions before and after data augmentation. Finally, the Label Me labeling tool was used to imitate the PASCAL VOC2012 dataset format to manually label the four categories of calculus, gingivitis, tartar, and wear surface in the lesion image and the dataset was divided into a training set, validation set, and test set at a ratio of 7:2:1.

The tartar data is small, but it has more prominent features such as color and texture. To cope with the small amount of tartar data, we incorporate some training techniques in the training process, such as pre-training the tartar dataset to initialize the weights and then using the pre-trained model to train our multi-class lesion segmentation model. We also oversample the tartar data during training by random replication, so that the prediction is more accurate for small amounts of tartar, and we add dropout to the network to prevent overfitting of the tartar data.

In addition, the tooth lesion dataset was overexposed, resulting in unclear edges and unclear details in the lesion image. The histogram of the normal tooth image is more balanced, while the overexposed tooth image has too many pixels with high brightness, which causes the histogram to shift to the right, as show in Figure 9.

This article uses the ACE [33] automatic color equalization algorithm to color balance the unclear images in the dataset. The algorithm considers the spatial positional relationship between color and brightness in the image, performs adaptive filtering of local characteristics, makes image brightness and color adjustments and contrast adjustments with local and nonlinear characteristics, and satisfies the gray world theory hypothesis and white speckle hypothesis. The ACE algorithm consists of two steps. The first step is to adjust the color/spatial domain of the image, complete the chromatic aberration correction of the image and obtain a spatially reconstructed image, as shown in Equation (3).
(3)$$R_{c}\left(p\right)=\underset{j=\mathrm{subset}}{\sum }\frac{r\left(I_{c}\left(p\right)-I_{c}\left(j\right)\right)}{d\left(p,j\right)}$$

In the formula, $R_{c}$ is the intermediate result, $I_{c}\left(p\right)-I_{c}\left(j\right)$ is the brightness difference between two different points, d(p, j) is the distance degree function, and r (*) is the degree performance function, which must be an odd function. This step can adapt to the local image contrast, r (*) can amplify small differences, enrich large differences and expand or compress the dynamic range according to the local content. It is generally agreed that r (*) is:(4)$$r\left(n\right)=\{\begin{array}{c} 1,\;\;\;\;\;\;\;\;\;\;\;\;\;\;\;\;\;\;X<T\\ X/T,-T<X<T\\ -1,\;\;\;\;\;\;\;\;\;\;\;\;\;\;\;X>T \end{array}$$

### 4.2. Metrics

In order to compare the performance of our method, we use the training model to segment the test set and compare the segmented image with the masked label. In terms of segmentation accuracy, our metrics mainly include pixel accuracy (Acc), average interaction ratio (mIoU), and F1 score. In terms of real-time performance, we mainly compare the amount of calculation and the reasoning time of a single picture. We perform semantic segmentation on tooth lesions, which is a pixel-level segmentation. In the field of deep learning image segmentation in computer vision, the mIoU value is an important metric to measure the accuracy of image segmentation. Assuming there are *k* + 1 classes, $P_{ij}$ represents the number of pixels whose actual class is class i but whose predicted result is class *j*. The calculation formula of mIoU is shown in Equation (5).
(5)$$\mathrm{MIou}=\frac{1}{K+1}\sum_{i=0}^{k}\frac{P_{ii}}{\sum_{i=0}^{k}p_{ij}+\sum_{i=0}^{k}p_{ji}-p_{ii}}$$

### 4.3. Loss Function

ICNet adds a loss weight to each branch training and optimizes the weighted SoftMax cross-entropy, and its loss function L can be expressed as:(6)$$L=W_{1}L_{1}+W_{2}L_{2}+W_{3}L_{3}$$
where $L_{1}$, $L_{2}$ and $L_{3}$ are the loss of the low-, medium-, and high-resolution branches, respectively, and $W_{1},{\;W}_{2}\;$and $W_{3}\;$are the weights of the loss function of the low-, medium-, and high-resolution branches, respectively. Normally, if the high-resolution branch weight $W_{1}$ is set to 1, the weights $W_{2}$ and $W_{3}$ of the medium-resolution and low-resolution branches are 0.4 and 0.16, respectively.

### 4.4. Experimental Details

We evaluated our method on a self-built dental lesion dataset. During the training process, we use Resnet50 as our backbone network and we use the SGD optimization method for training when the loss of the validation set does not decrease in 20 epochs. The data transformation of loss and acc in the training process is shown in Figure 10. Our method has a faster convergence rate. We replace the large-size convolution in the spatial attention module in CBAM and perform ablation experiments. Finally, we enhanced and reduced the brightness of the original image and tested the segmentation results of our model under different brightness levels.

The experiments in this article are all based on the TensorFlow deep learning framework, completed on the Bit hub cloud server, and the graphics card information is gtx1080.

## 5. Results

### 5.1. Contrast Test with Other Segmentation Algorithms

At present, with the development of convolutional neural networks, an increasing number of deep learning methods are used for semantic image segmentation. However, the segmentation performance of different tasks and methods is significantly different. To further test the pros and cons of this method for real-time semantic segmentation of dental lesion images, the training parameters of the above four models are the same as those of the improved ICNet. They are all trained based on the strategy of automatically saving the optimal model and then tested on the verification set. The test indicators mainly start from the two aspects of segmentation accuracy and time performance and include Acc, mIoU, the F1 score, and the reasoning time of a single picture. It can be seen from Table 2 that our method has the highest Acc, mIoU, and F1 score, with scores of 0.8897, 78.67%, and 0.8890, respectively. Compared with ICNet, the improved ICNet segmentation accuracy, mIoU, and F1 scores are 0.0384, 3.91%, and 0.0397 higher, respectively.

In addition, from the results of the visualization in Figure 11, U-Net and SegNet have oversegmentations. That is, the no lesion part is segmented, while ENet and ICNet have undersegmentations, which makes it difficult to correctly identify normal tooth tissues and lesions. The improved ICNet greatly improves the oversegmentation, and the result is closer to the label.

In terms of real-time segmentation, ENet, UNet, SegNet, ICNet, and the improved ICNet took 833 ms, 696 ms, 739 ms, 805 ms, 307 ms, and 395 ms for a single image, respectively. Compared with ENet, U-Net, and SegNet, the improved ICNet shortens the time by 34.63%, 38.43%, and 43.47%, respectively, and shortens the time by nearly half compared with FCN and increases less time compared with ICNet. From Table 3, we can see that the improved ICNet is slightly more computationally expensive than ICNet.

From the above evaluation indicators, it can be seen that our method has the best segmentation effect, and when the segmentation time is close to ICNet, the three indicators of our method have improved to varying degrees. The indicators have improved to varying degrees. Among the visualization results of all methods, our segmentation effect is the closest to the real label. The abscissa of Figure 12 is the frequency of division, the ordinate is MIOU, and the upper-right corner of the figure is the optimal method. Figure 12 shows that our method is optimal in terms of accuracy and time.

Among the four types of lesions, dental calculus and wear surface are the most similar to normal tooth tissue characteristics. Figure 13 shows the segmentation visualization results of dental calculus and wear surface by our method and other algorithms. FCN, U-Net, and SegNet oversegment for calculus. ENet and ICNet have certain undersegmentation problems. The visualization effect of our method is closest to the label. Among the visualization results of wear surface segmentation, ENet segmentation has the worst effect, the segmentation results of the other algorithms are not much different, and the results are relatively close.

### 5.2. Segmentation under Different Brightness

When an oral image is collected, the brightness of each image is difficult to maintain, and the brightness becomes one of the important factors that affects the quality of the segmentation. Therefore, this article compares the segmentation and dental lesion images under different brightness levels. We set the original image brightness to P, reduce the image brightness to 0.7 times that of the original image, and increase it to 1.3 times that of the original image for image segmentation.

It can be seen from Table 4 that when the brightness is adjusted to 0.7 P or 1.3 P, the segmentation accuracy is reduced. The analysis believes that it is more difficult to identify the characteristics of the lesion in the image that is too bright and too dark. The ICNet segmentation accuracy dropped by 2.33% and 4.5%, respectively, and our method’s segmentation accuracy dropped by 2.68% and 5.34%, respectively. According to the visualization results, in Figure 14, ICNet has an undersegmentation at 0.7 P and an oversegmentation at 1.3 P. Our method is closer to the real label at 0.7 and 1.3 P.

### 5.3. Ablation Experiment of CBAM

We replace the 7 × 7 large-size convolution kernel in the spatial attention in CBAM with a convolution kernel with a size of 3 × 3 and a dilated rate of 2. As shown in Table 4, the standard convolution with a convolution kernel size of 7 × 7 is represented by ICnet + CBAM_7 × 7_; the convolution kernel size is 3 × 3, and the void convolution with the void ratio is 2, using ICNet + CBAM_3 × 3_ means. We use a dilated convolution with a convolution kernel size of 3 × 3 and a dilated rate of 2 twice, denoted by CBAM_2 × 3 × 3_. Compared with the parameter quantity of ICNet + CBAM_3 × 3_ as the benchmark, we set the parameter increment of ICNet + CBAM_3 × 3_ to 0.

From Table 5, we can see that after replacing the 7 × 7 convolution in the spatial attention with two 3 × 3 convolutions, the mIoU increased by 0.55% when the number of parameters was reduced. Experiments show that a 3 × 3 dilated convolution with a dilated rate of 2 has the same perceptual field as a standard convolution of 7 × 7, and the effect of both is approximate. In addition, comparing the results of the third group of experiments, it is found that stacking multiple convolutional layers with dilated layers can also improve the feature expression abilities of the spatial attention module.

## 6. Conclusions

In this paper, for a small sample of dental lesion datasets, the data are augmented by random cropping and flipping. The ACE automatic color equalization algorithm addresses the problems of blurred images of the lesion caused by the light source. Combining the augmented data and processed image with the original dataset into a new dataset makes the resulting model more generalizable.

The edge of the tooth lesion is highly similar to the normal tooth, and the edge is difficult to subdivide. Moreover, although ICNet obtains most of the semantic information when performing feature extraction in low- and medium-resolution branches, the details are easily lost, resulting in inaccurate lesion segmentations. Therefore, this article adds a lightweight CBAM module to the feature extraction stage, which can better obtain the semantic information of the image so that the high-resolution branch can better guide the low-resolution-generated feature map, thereby improving the segmentation accuracy.

We replace the 3 × 3 convolution with an asymmetric convolution in the low- and medium-resolution convolution stages, which further reduces the computational complexity of the model. We also ensure that the time it takes to improve the accuracy of the model does not increase significantly.

Although our method has significantly improved the segmentation of a variety of dental lesions, the light intensity has a large impact on the segmentation effect. In addition, the labeling of lesion images requires a great deal of effort. In future research, we will establish a highly generalized, weakly supervised network to solve the problems of difficult labeling and the large impact of the light.

## Figures and Tables

**Figure 1 micromachines-13-01920-f001:**
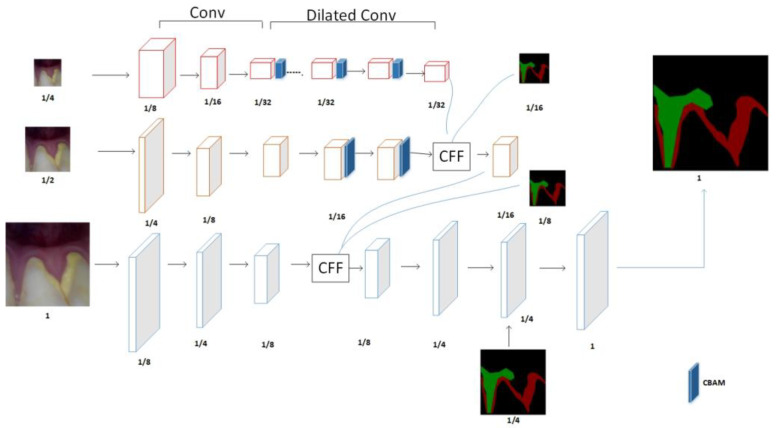
Improved ICNet network structure.

**Figure 2 micromachines-13-01920-f002:**
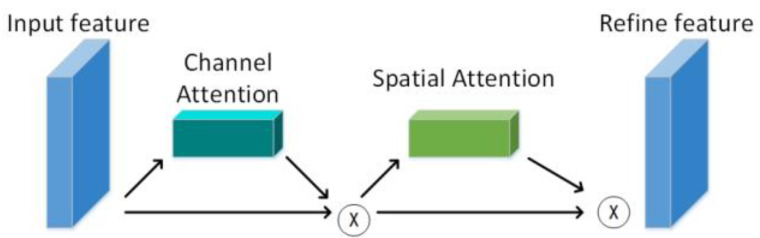
The serial structure of channel attention and spatial attention.

**Figure 3 micromachines-13-01920-f003:**
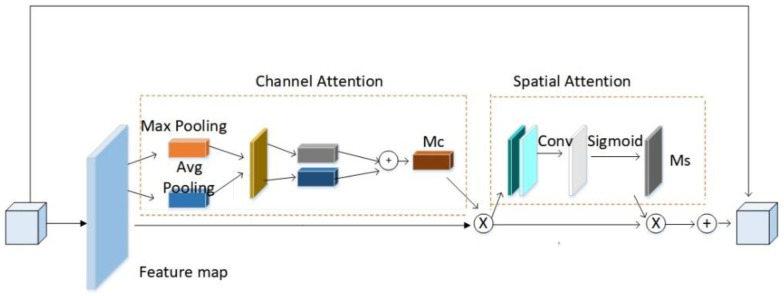
Embedding CBAM outside the convolution block.

**Figure 4 micromachines-13-01920-f004:**
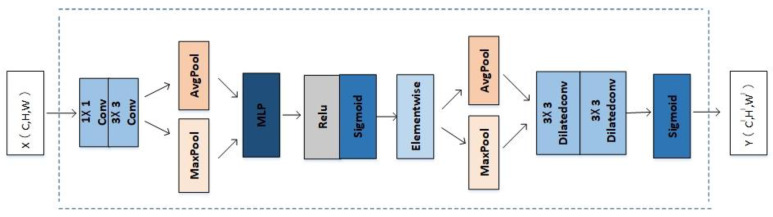
Use stacked dilated convolution to replace the spatial attention structure diagram of large-size convolution.

**Figure 5 micromachines-13-01920-f005:**
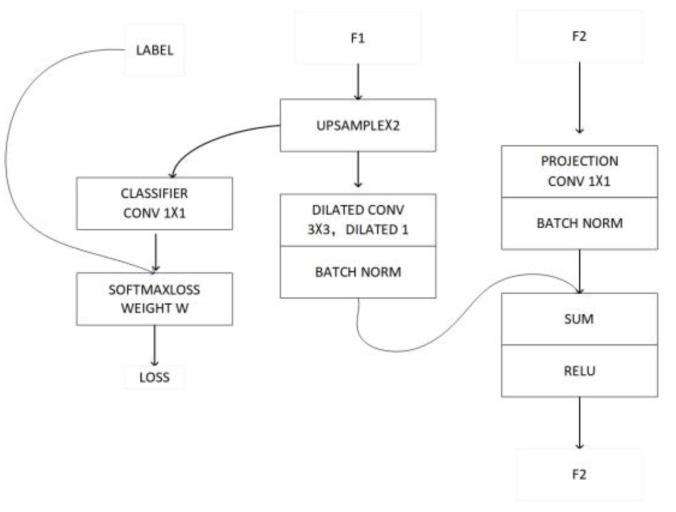
Feature maps F1 and F2 are fused through CFF.

**Figure 6 micromachines-13-01920-f006:**
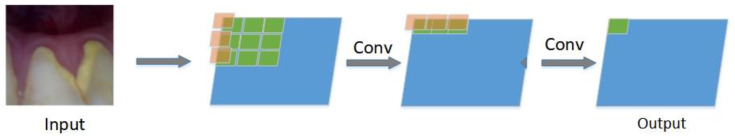
Asymmetric convolution process.

**Figure 7 micromachines-13-01920-f007:**
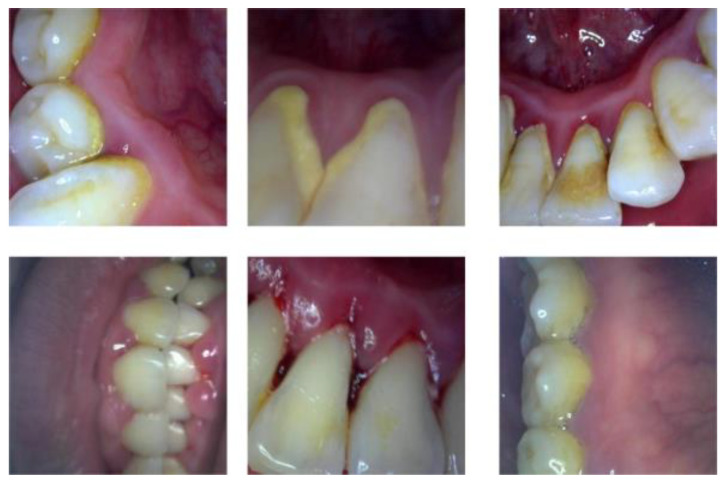
Example of combined dataset.

**Figure 8 micromachines-13-01920-f008:**
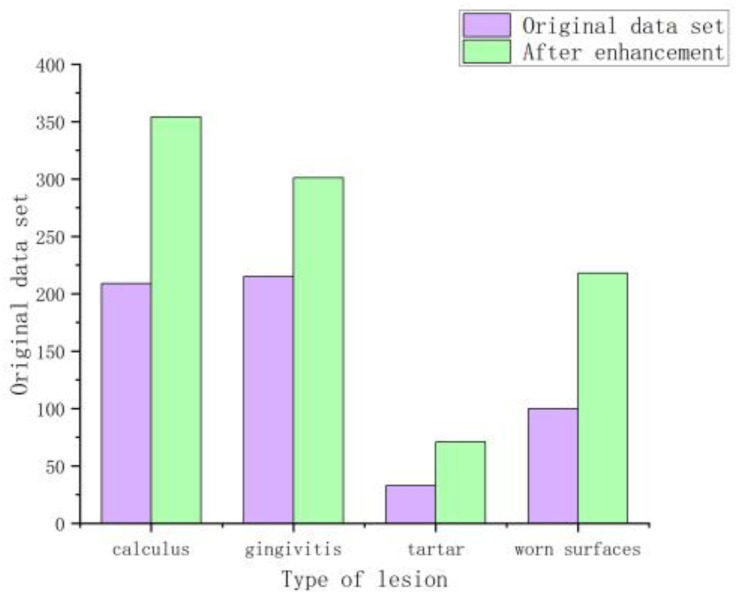
Comparison of the number of occurrences of each lesion on the original dataset and the combined dataset.

**Figure 9 micromachines-13-01920-f009:**
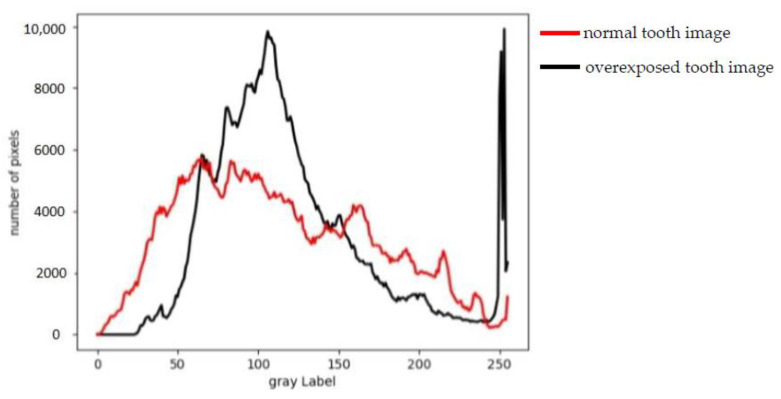
Comparison of gray histogram between a normal tooth image and an overexposed tooth image.

**Figure 10 micromachines-13-01920-f010:**
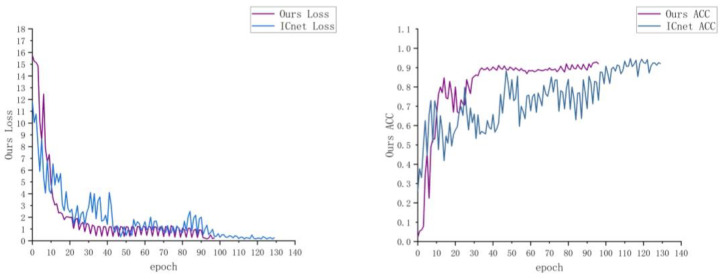
Comparison of loss and ACC curve.

**Figure 11 micromachines-13-01920-f011:**
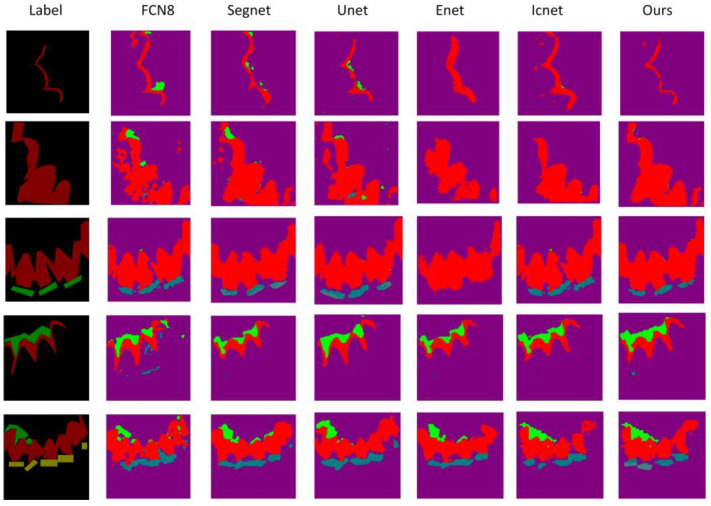
Visual comparison chart of segmentation results of different algorithms.

**Figure 12 micromachines-13-01920-f012:**
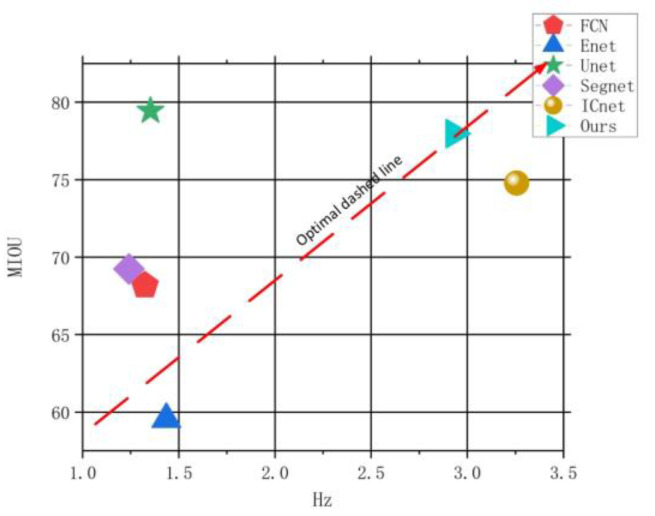
Comparison of accuracy and efficiency of different segmentation algorithms.

**Figure 13 micromachines-13-01920-f013:**
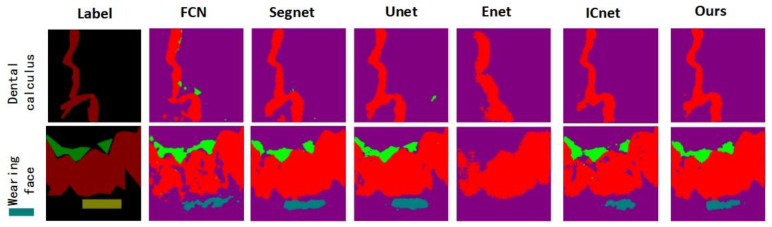
Visual comparison of segmentation results of dental calculus and worn surface.

**Figure 14 micromachines-13-01920-f014:**
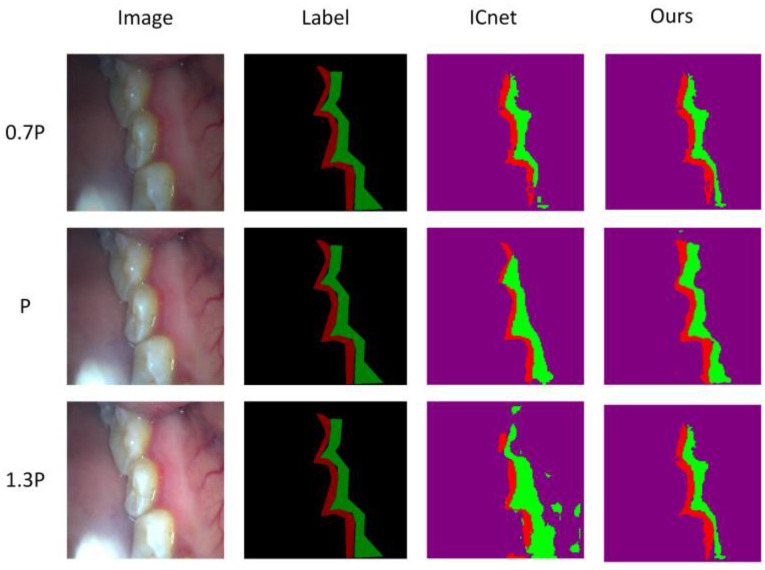
Visualization comparison of segmentation results under different brightness.

**Table 1 micromachines-13-01920-t001:** The number of times that various lesions appear on the collected images.

Calculus	Gingivitis	Tartar	Worn Surfaces
209	215	33	100

**Table 2 micromachines-13-01920-t002:** Comparison of various segmentation indices of different algorithms.

Model	Acc	mIoU	F1_Score	Times (ms)
FCN8	0.8215	68.17	0.8045	833
ENet	0.8160	62.30	0.7749	696
U-Net	0.8875	78.61	0.8838	739
SegNet	0.8284	69.24	0.8162	805
ICNet	0.8513	74.76	0.8493	**307**
Ours	**0.8897**	**78.67**	**0.8890**	395

**Table 3 micromachines-13-01920-t003:** Comparison of calculations before and after ICNet improvement.

Model	FLOPs
ICNet	13,524,726
Ours	15,608,042

**Table 4 micromachines-13-01920-t004:** mIoU comparison of our method with the ICNet method at different luminance.

mIoU			
Model	0.7P	P	0.8P
ICNet	58.50	60.83	56.38
Ours	58.65	61.33	55.99

**Table 5 micromachines-13-01920-t005:** Replace the 7 × 7 convolution in the spatial attention in CBAM with a different convolution.

Model	MIOU	Add Param
ICNet + CBAM_3 × 3_	76.84	0
ICNet + CBAM_7 × 7_	77.42	240
ICNet + CBAM_2 × 3 × 3_	77.97	180

## Data Availability

Not applicable.

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
