# Peer review of "Dental Lesion Segmentation Using an Improved ICNet Network with Attention"

_micromachines, 2022, doi:10.3390/mi13111920_

Round 1

Reviewer 1 Report

This manuscript proposes an improved ICNet model for dental lesions segmentation. Through integrating a CBAM attention module and partly replacing the convolution in the network, the method improves the segmentation effect of various dental lesions in an accurate and efficient manner. I recommend that this well-organized paper be accepted after addressing some concerns as follows.

1. On page 6 "α is the activation function", where is the α in the Formula (1) and (2)? Besides, there are two formulas numbered “(1)” on page 6, please correct the equation number.

2. I suggest that the authors give a brief description of Figure 5 and the CFF module used in this paper.

3. The dataset for tartar is much smaller than the other three classes. I am curious if it will influence the performance evaluation of the segmentation model, such as overfitting.

4. What does “A-ICnet” stand for in Figure 10? Better demonstrations should be added to make the comparison clear.

5. Some formatting issues of references should be improved with the journal standard. For example, ref 1, 6, and 14 use different formats.

Reviewer 2 Report

1. The research is meaningful to clinical applications. The work itself is of good quality with potential good interest to the community.

2. This paper is dealing with segmentation of tooth lesions for tooth lesion detection. ICNet network is proposed to segment various lesion types in real-time. Experimental results show the solution is promising. 

3. Segmentation is a hot research topics in AI. Some state of the art should be added in the literature review:

1) 3D semantic segmentation with submanifold sparse convolutional networks.

   CVPR 2018 Open Access Repository (thecvf.com)

2) BEACon: a boundary embedded attentional convolution network for point cloud instance segmentation.

   DOI https://doi.org/10.1007/s00371-021-02112-7

3) Three dimensional imaging techniques: A literature review

   doi: 10.4103/1305-7456.126269

4. The article is not a review type

5. Introduction should not be Section 0.

Reviewer 3 Report

The manuscript discusses a dental lesions segmentation method based on the improved ICNet. Tooth lesions are similar to normal tooth tissues, and the method integrated the CBAM attention module into the ICNet network structure, and replaced the large-size convolutions in the spatial attention module with layered dilated convolutions to enhance the relevant features. This work is interesting, and helpful for intelligent dental diagnosis. So, the manuscript is suggested to be accepted. However, there are still some suggestions for modification:

1) The review part should contain the papers from the year 2021, and the focus should be on a critical analysis of the gradual advancement, as well as the current level, of the state-of-the-art, with quantitative information on the time & space complexity.

2) Each shape in Figure 1 should have a label, just like the one in the right-bottom corner. And, it is suggested to give a detailed description of Figure 1 in the text.

3) Seen from table 1, the number of different types of lesions is distributed unevenly. Does this affect the algorithm training? It is suggested to give a explanation.
